# Evaluation of a FlpA Glycoconjugate Vaccine with Ten *N*-Heptasaccharide Glycan Moieties to reduce *Campylobacter jejuni* Colonisation in Chickens

**DOI:** 10.3390/vaccines12040395

**Published:** 2024-04-09

**Authors:** Ricardo Corona-Torres, Prerna Vohra, Cosmin Chintoan-Uta, Abi Bremner, Vanessa S. Terra, Marta Mauri, Jon Cuccui, Lonneke Vervelde, Brendan W. Wren, Mark P. Stevens

**Affiliations:** 1The Roslin Institute and Royal (Dick) School of Veterinary Studies, University of Edinburgh, Easter Bush, Edinburgh EH25 9RG, UK; rcorona@exseed.ed.ac.uk (R.C.-T.); prerna.vohra@ed.ac.uk (P.V.); cosmin.chintoan@outlook.com (C.C.-U.); a.bremner.07@aberdeen.ac.uk (A.B.); lonneke.vervelde@roslin.ed.ac.uk (L.V.); 2Institute for Immunology and Infection Research, School of Biological Sciences, Charlotte Auerbach Road, University of Edinburgh, Edinburgh EH9 3FF, UK; 3Department of Infection Biology, London School of Hygiene and Tropical Medicine, Keppel Street, London WC1E 7HT, UK; vanessa.terra@lshtm.ac.uk (V.S.T.); marta.mauri@lshtm.ac.uk (M.M.); jon.cuccui@lshtm.ac.uk (J.C.); brendan.wren@lshtm.ac.uk (B.W.W.)

**Keywords:** *Campylobacter*, poultry, vaccine, FlpA, heptasaccharide, glycoconjugate

## Abstract

*Campylobacter* is a major cause of acute gastroenteritis in humans, and infections can be followed by inflammatory neuropathies and other sequelae. Handling or consumption of poultry meat is the primary risk factor for human campylobacteriosis, and *C. jejuni* remains highly prevalent in retail chicken in many countries. Control of *Campylobacter* in the avian reservoir is expected to limit the incidence of human disease. Toward this aim, we evaluated a glycoconjugate vaccine comprising the fibronectin-binding adhesin FlpA conjugated to up to ten moieties of the conserved *N*-linked heptasaccharide glycan of *C. jejuni* or with FlpA alone. The glycan dose significantly exceeded previous trials using FlpA with two *N*-glycan moieties. Vaccinated birds were challenged with *C. jejuni* orally or by exposure to seeder-birds colonised by *C. jejuni* to mimic natural transmission. No protection against caecal colonisation was observed with FlpA or the FlpA glycoconjugate vaccine. FlpA-specific antibody responses were significantly induced in vaccinated birds at the point of challenge relative to mock-vaccinated birds. A slight but significant antibody response to the *N*-glycan was detected after vaccination with FlpA-10×GT and challenge. As other laboratories have reported protection against *Campylobacter* with FlpA and glycoconjugate vaccines in chickens, our data indicate that vaccine-mediated immunity may be sensitive to host- or study-specific variables.

## 1. Introduction

*Campylobacter* is the leading bacterial cause of foodborne illness and was estimated to have caused 95.6 million infections, 21.3 thousand deaths, and the loss of 2.1 million disability-adjusted life years in 2010 [1]. In the United Kingdom, an estimated 500,000 cases of human campylobacteriosis occur each year [2], at a cost to the healthcare system of GBP 50 million per year [3]. Infections in humans typically involve acute gastroenteritis, which can be followed by inflammatory neuropathies, reactive arthritis, and other sequelae. Source attribution studies unequivocally show that poultry are a key reservoir of human infections, with the handling or consumption of chicken meat being a key risk factor [4]. Annual surveys of fresh retail chicken across the United Kingdom from 2014–2019 have found between 54% and 77% of neck skin samples to be contaminated with *Campylobacter*, with 6–24% of samples having over 1000 colony-forming units (CFU) per gram [5].

*Campylobacter* can reach high levels in the intestines of chickens, often in the absence of overt clinical signs. However, in some broiler lines, *C. jejuni* elicits intestinal inflammation [6] and can impair intestinal barrier function, nutrient uptake, and growth [7,8]. Strategies for the control of *Campylobacter* in poultry therefore offer the potential to reduce zoonotic infections and improve the productivity and welfare of some broiler lines. Modelling and quantitative microbiological risk assessments indicate that a 100-fold reduction in caecal colonisation of chickens could reduce human infections due to contaminated broiler meat by 30-fold [9] or 76–98% [10]. Towards this aim, we and others have evaluated live-attenuated *Salmonella* vaccines vectoring antigens from *C. jejuni*, subunit vaccines based on *C. jejuni* proteins, live-attenuated *E. coli* or outer membrane vesicles decorated with the conserved *N*-heptasaccharide glycan of *C. jejuni*, and glycoconjugate vaccines comprising the heptasaccharide glycan *N*-linked to *Campylobacter* proteins or detoxified carrier proteins.

The inexpensive production of glycoconjugate vaccines has been enabled by the expression of the glycan of interest in laboratory-adapted *E. coli* together with the *Campylobacter* oligosaccharyltransferase PglB, which couples glycans to proteins containing the motif D/E-X1-N-X2-S/T (where X1 and X2 can be any amino acid except proline). These motifs can be introduced into a protein antigen from the target pathogen or a protein with adjuvant properties, which are then targeted to the bacterial periplasm where glycosylation occurs and affinity purified (e.g., by inclusion of a 6 × histidine tag to enable capture on cobalt resin). This strategy has been used to develop promising glycoconjugate vaccines against, *Burkholderia pseudomallei* [11], *Staphylococcus aureus* [12], and *Shigella flexneri* [13].

The *N*-linked heptasaccharide of *C. jejuni* is an appealing vaccine candidate as it is highly conserved in *Campylobacter* species and plays a key role in colonisation of the avian host. Glycoconjugate vaccines for *Campylobacter* in poultry have been reported to offer variable protection. Nothaft et al fused a detoxified version of ToxC from *Corynebacterium diphtheriae* to a naturally occurring peptide from *C. jejuni* containing nine repeats of the glycosylation motif. Vaccinated chickens had caecal colonisation levels of 3 to 9 log_10_ colony-forming units (CFU)/g, compared to 10 to 11 log_10_ CFU/g in the control group [14]. However, independent studies with a detoxified version of ExoA from *P. aeruginosa* with up to ten moieties of the *C. jejuni* heptasaccharide did not confer protection against caecal colonisation [15]. Similarly, outer membrane vesicles from *E. coli* decorated with the *C. jejuni N*-heptasaccharide have been reported to be protective in some studies [16] but not others [17].

We previously evaluated a glycoconjugate vaccine based on *C. jejuni* fibronectin-binding adhesin FlpA with up to two *N*-heptasaccharide moieties [18]. While FlpA has been reported to elicit protection against *C. jejuni* in chickens [19], we observed that neither FlpA nor glycosylated FlpA conferred protection against oral challenge with *C. jejuni* M1 [18]. Here, we evaluated a version of FlpA with up to ten *N*-heptasaccharide moieties to determine if a greater *N*-glycan dose is necessary for protection. Increasingly, the number of glycan moieties per carrier protein has been reported to enhance protection by a glycoconjugate vaccine against *F. tularensis* [20]. We also hypothesised that vaccine-induced immune responses may have been overwhelmed by the challenge strain used in previous studies (M1), which can rapidly colonise chickens to reach over 9 log_10_ CFU/g in the caeca within just 7 days of inoculation with just 100 CFU. We therefore used strain NCTC 11168H, which requires a minimum dose of 4 log_10_ CFU to reliably establish colonisation and does so at a slower rate than M1 [15]. Additionally, we developed and used a seeder-bird challenge model, whereby chickens pre-colonised with *C. jejuni* are introduced to a population of vaccinated birds to mimic natural transmission.

## 2. Materials and Methods

### 2.1. Design of a Construct for Expression of FlpA with up to 10 N-Heptasaccharide Glycan Moieties (FlpA-10×GT)

The coding sequence for the *C. jejuni* FlpA gene was modified to insert five repeats encoding the glycosylation site DQNAT at each terminus of the gene. Additionally, a ribosome binding site and PelB signal sequence were added at the N-terminus, and a 6 × histidine tag was introduced at the C-terminus, essentially as described [18]. The construct was codon optimised for expression in *E. coli*, commercially synthesised (Thermo Fisher, Waltham, MA, USA), and cloned via *Xba*I and *Hind*III restriction sites into pFPV25.1 [21], under the control of a constitutive *rpsM* promoter. The resulting plasmid (pFlpA-10×GT) was transformed into *E. coli* strain χ7122*pgl*, which has the *C. jejuni pgl* locus stably inserted in the bacterial chromosome [22]. Transformants were cultured on Luria Bertani agar with 100 µg ml^−1^ ampicillin (for pFlpA-10×GT) and 50 µg mL^−1^ kanamycin (for strain χ7122*pgl*). To produce the control unglycosylated FlpA, the wild-type *flpA* gene was amplified by PCR from the genomic DNA of *C. jejuni* strain M1 and cloned into pET26b (Novagen, Madison, WI, USA) under the control of a T7 promoter with a C-terminal 6 × histidine tag via *Nde*I and *Xho*I restriction endonuclease sites to make pET26b-*flpA*. FlpA with up to 2 *N*-glycan moieties (FlpA-2×GT) was purified as described previously [18].

### 2.2. Protein Expression and Purification

Histidine-tagged NetB and NetB-10×GT used for enzyme-linked immunosorbent assays were purified as previously described [15]. Strain χ7122*pgl* pFlpA-10×GT was cultured in 2 L of Luria Bertani broth containing antibiotics as above for 24 h at 30 °C. To express wild-type FlpA, *E. coli* BL21(DE3) containing pET26b-*flpA* was grown at 37 °C until the cell density reached an optical density of 0.6 at 600 nm. Protein expression was induced with 0.1 mM IPTG, and the culture was incubated for 20 h at 30 °C. In both cases, cells were collected by centrifugation and disrupted with a One Shot cell disrupter (Constant Systems, Daventry, UK). Bacterial lysates were filter-sterilised and the proteins purified by cobalt affinity chromatography on a 1 mL Hi-Trap TALON crude column (Cytiva, Little Chalfont, UK). The protein was eluted in an imidazole solution (50 mM sodium phosphate, 300 mM NaCl, 150 mM imidazole, pH 7.4) according to the manufacturer’s instructions. Eluted fractions containing the protein were identified by sodium dodecyl sulphate-polyacrylamide gel electrophoresis (SDS-PAGE), combined, and dialysed in 20 mM Tris, pH 8.0, using a 10 KDa molecular weight cut-off Slide-A-Lyzer cassette (Thermo Fisher, Waltham, MA, USA). The purity of FlpA and FlpA-10GT was assessed by Coomassie staining of proteins resolved by SDS-PAGE and Western blotting using a mouse monoclonal anti-6×His 680LT conjugate (Li-COR, Cambridge, UK) at 1:10,000 to detect the protein and biotinylated soybean agglutinin (SBA) at 1:400 (Vector Laboratories, Newark, CA, USA) together with streptavidin IRDye 800CW at 1:4000 (Li-COR, Cambridge, UK) to detect glycosylated protein. Blotted membranes were imaged with an Odyssey Fc Imager (Li-COR, Cambridge, UK) using the 700 and 800 filters to detect bound anti-6×xHis and SBA, respectively. Protein concentration was determined by NanoDrop (ThermoFisher, Waltham, MA, USA), using the extinction coefficient determined for the modified protein calculated using Prot Pram (Expasy): FlpA extinction coefficient 54,780 M^−1^ cm^−1^, Abs (0.1%) = 1.25, Molecular weight (MW) = 49,042.09 Da. Densitometry was performed using the Li-COR software Image Studio version 5.2.5 to estimate the efficiency of glycosylation by comparing the 6×His signal of bands where no *N*-glycan was detected to that where they were. The proportions of each glycosylated band were also determined from individual SBA signal intensities. The maximum amount of glycan on each band was calculated from the molecular weights of FlpA and *N*-glycan (for example, FlpA with 1 glycan, FlpA with 2 glycans, etc.) and multiplied by the glycosylation efficiency to determine the *N*-glycan/FlpA ratio at each band and the total amount of *N*-glycan in the sample.

### 2.3. Culture of Campylobacter jejuni

To prepare challenge inocula, *C. jejuni* NCTC 11168H was cultured on charcoal cephoperazone deoxycholate agar (CCDA; Oxoid, Basingstoke, UK) from cryopreserved single-use aliquots for 48 h. Colonies were transferred to tubes containing Mueller-Hinton (MH) broth (Oxoid, Basingstoke, UK) and incubated with shaking overnight to achieve the stationary phase of growth. All the cultures were incubated in an anaerobic workstation (Don Whitley Scientific, Sheffield, UK) at 41 °C under microaerophilic conditions (5% O_2_, 5% CO_2_, and 90% N_2_). Bacterial motility was assessed visually, and a culture containing at least 80% motile *C. jejuni* cells was diluted in phosphate-buffered saline (PBS) to the required concentration to inoculate the birds as described [15], and viable counts were retrospectively confirmed by plating serial ten-fold dilutions to CCDA and enumerating colonies after 48 h. To quantify bacterial colonisation, equal amounts of contents from each caecum were pooled, diluted serially tenfold in sterile PBS, and plated on CCDA, followed by incubation for 48 h.

### 2.4. Development of a Challenge Model to Mimic Natural Transmission of C. jejuni

Animal experiments were conducted at the Moredun Research Institute according to the requirements of the Animals (Scientific Procedures) Act 1986 under project licence PCD70CB48 with the approval of the local Animal Welfare and Ethical Review Board. White Leghorn chickens free of *Campylobacter* were supplied by the National Avian Research Facility, University of Edinburgh, and were provided access to irradiated feed based on vegetable protein (DBM Ltd., Broxburn, UK) and water ad libitum. Groups were of mixed sex, and individuals were wing-tagged for identification. Seven birds were inoculated with 10^8^ CFU/bird of *C. jejuni* NCTC 11168H by oral gavage at 14 days post-hatch. On day 21 post-hatch (7 days post-inoculation), two of these ‘seeder’ birds were introduced into a group of 23 ‘sentinel’ birds aged 21 days old held in a separate room to prevent cross-contamination. The remaining 5 seeder-birds were humanely euthanised by a Schedule 1 method in order to enumerate caecal *C. jejuni* as described above. Caecal colonisation of the sentinel birds was determined 7 and 10 days after they were placed in contact with seeder-birds, with 11 and 12 birds being sampled at these intervals, respectively. Caecal colonisation of the 2 seeder-birds was analysed over a 7-day time interval.

### 2.5. Trials to Evaluate the Efficacy of FlpA and FlpA-10×GT as Subunit Vaccines against C. jejuni

Vaccines were prepared by mixing purified unglycosylated or glycosylated FlpA with TiterMax Gold adjuvant (Merck, Glasgow, UK) at a 1:1 ratio on the days of inoculation. Each dose contained 100 μg of FlpA protein or the molar equivalent of FlpA-10×GT in 100 µL. A total of 138 White Leghorn birds were divided equally into 6 groups of 23 birds and vaccinated via the subcutaneous route on the day of hatch and on day 14 post-hatch, with 50 μL being injected into each side of the thorax. Birds in groups 1 and 4 were vaccinated with FlpA; groups 2 and 5 were vaccinated with FlpA-10×GT; and groups 3 and 6 were mock-vaccinated with 100 μL of PBS and TitreMax Gold adjuvant at a 1:1 ratio.

Two weeks after the second vaccination (aged 28 days old), birds from groups 1 to 3 were challenged with 10^5^ CFU/bird of *C. jejuni* NCTC 11168H by oral gavage. A dose of 10^4^ CFU of this strain has previously been reported to result in reliable caecal colonisation in chickens of this age, but at a slower rate than observed for strain M1 [15]. Groups 4 to 6 were challenged by introducing 2 seeder-birds per group that had been previously infected with *C. jejuni* NCTC 11168H, as described above. Vaccinated and mock-vaccinated groups exposed to seeder-birds were housed separately. Blood samples were collected from the brachial veins of 5 birds per group 2 weeks after each vaccination. Postmortem examinations were performed 7 and 28 days post-challenge. Owing to spontaneous mortality within the normal range for the line and age of chickens used, the number of birds per time point was variable. Between 9 and 11 birds from each group were euthanised 7 days post-challenge, with the remaining birds being euthanised 28 days post-challenge. Blood was collected by cardiac puncture at postmortem examination, centrifuged after clotting at 1000× *g* for 10 min at 4 °C, and the resulting serum was then stored at −80 °C. The contents from both caeca of each bird were pooled and 10-fold serial dilutions prepared in PBS and plated on CCDA to determine viable counts of *C. jejuni* per gram.

### 2.6. Analysis of Vaccine-Induced Humoral Immune Responses

Enzyme-linked immunosorbent assays (ELISA) were used to quantify antigen-specific serum Ig levels. Briefly, 96-well MaxiSorp plates (Merck, Glasgow, UK) were coated with 0.1 µg/well of the FlpA-10×GT antigen or the cognate FlpA unglycosylated antigen to quantify FlpA-specific responses. Plates were coated using carbonate–bicarbonate buffer at 4 °C overnight, then washed with PBS containing 0.005% (*v*/*v*) Tween-20. Serum samples were diluted in PBS, and 100 µL of diluted serum was added per well. Serum dilutions of 1:50 were selected as described for FlpA and FlpA-2×GT in preceding studies [18]. Control wells were used, to which no serum was added. To confirm the specificity of the Ig detected (i.e., control for contaminants co-purified from *E. coli*) and attempt to detect *N*-glycan-specific antibodies, serum from chickens vaccinated with FlpA-10×GT or FlpA was tested against 6 × His-tagged NetB or NetB-10×GT as described [18] and vice versa. Plates were incubated at 37 °C for 1 h and then washed as above. To detect bound serum antibody, goat anti-chicken IgY-horseradish peroxidase (HRP) was added at 1:3000 (ab6753, Abcam, Cambridge, UK), with incubation for a further 1 h followed by washing twice. The tetramethylbenzidine substrate (BioLegend, London, UK) was then added, and the plates were incubated for 10 min at room temperature in the dark. The reaction was stopped using 2 M H_2_SO_4_, and absorbance at 450 nm adjusted against absorbance at 620 nm (A_450_/A_620_) was measured using a Biotek Cytation 3 plate reader (Agilent Technologies, Santa Clara, CA, USA) with background correction using the values of the control wells.

### 2.7. Statistical Analysis

GraphPad Prism version 8.00 (GraphPad Software, San Diego, CA, USA) was used to perform statistical analysis. The distribution of the data was evaluated by using the D’Agostino-Pearson and Shapiro-Wilk normality tests. A Kruskal-Wallis test followed by Dunn’s multiple comparison test was used to analyse differences in colonisation levels and humoral responses between groups of vaccinated chickens at each time point. The data are represented graphically as median values with 95% confidence intervals. *p* values ≤ 0.05 were considered to be statistically significant.

## 3. Results

### 3.1. FlpA-10×GT Is N-Glycosylated to a Substantially Greater Extent Than FlpA-2×GT

FlpA, FlpA-2×GT, and FlpA-10×GT were affinity purified as described in the Materials and Methods and analysed by SDS-PAGE and Western blotting. Unglycosylated proteins of the expected sizes were detected, accounting for variation in the number of DQNAT glycosylation sites added (Figure 1A). As previously [18], variants of FlpA-2×GT with up to two *N*-glycan moieties were observed following Western blotting with biotinylated soybean agglutinin detected with a fluorophore-streptavidin conjugate (Figure 1B). For FlpA-10×GT, variants with up to eight *N*-glycan moieties were readily observed within the limit of detection of the assay. Using densitometry, glycosylation efficiency was calculated to be 81%, and each vaccine dose of 100 μg FlpA-10×GT was estimated to contain 9.98 μg of glycan (see Appendix A), significantly greater than we previously achieved with FlpA-2×GT (3 μg, [18]).

### 3.2. Development of a Challenge Model to Mimic Natural Transmission of C. jejuni

In previous studies, we have titrated the dose of *C. jejuni* strains such that vaccinated chickens are given the minimum dose required to reliably establish caecal colonisation [15,17,18]. To refine our challenge protocol to mimic natural transmission, we developed a model where naïve sentinel birds are co-housed with ‘seeder’ birds previously inoculated with 10^5^ CFU *C. jejuni* NCTC 11168H (Figure 2A). Seven birds were inoculated by oral gavage at 14 days of age, and one week later they were colonised at c. 10 log_10_ CFU of *C. jejuni* per gram of caecal contents with little bird-to-bird variation (Figure 2B). Two of these birds were introduced as seeder-birds into a group of 23 sentinel chickens housed in a separate room at 21 days of age, while the other 5 birds were euthanised to quantify the caecal colonisation of the orally inoculated birds on the day of challenge (Figure 2B). At 7 and 10 days after the introduction of the seeder-birds, 11 and 12 birds were respectively euthanised and found to contain c. 9.5 log_10_ CFU/g *C. jejuni* in the caecal content. The two seeder-birds were euthanised 7 days post-challenge, and caecal *C. jejuni* counts showed that they were colonised at comparable levels (Figure 2B). No significant difference between colonisation levels one week after oral gavage with 10^5^ CFU of NCTC 11168H or in-contact exposure was observed.

### 3.3. Vaccination of Chickens with FlpA or FlpA-10×GT Does Not Significantly Reduce Caecal C. jejuni Colonisation

Groups of 23 chickens were vaccinated subcutaneously with FlpA, FlpA-10×GT, or PBS 1:1 in TitreMax Gold adjuvant and then challenged either by oral gavage with 10^5^ CFU NCTC 11168H or by in-contact exposure to pre-colonised seeder-birds as described above (Figure 3A). Following oral inoculation, we observed an unexpectedly high inter-animal variance in caecal *C. jejuni* levels in the mock-vaccinated group at 7 days post-challenge, with a median of c. 5.5 log_10_ CFU/g (Figure 3B). By 28 days after oral challenge, caecal colonisation levels were less variable in the mock-vaccinated birds, with a median of c. 7.7 log^10^ CFU/g. In chickens vaccinated with FlpA or FlpA-10×GT, caecal *C. jejuni* levels were higher than in the mock-vaccinated group 7 days after oral challenge (significantly so in the case of the FlpA group; *p* = 0.025). No significant differences were detected between the groups at 28 days after oral challenge (Figure 3B). The median colonisation level in chickens vaccinated with FlpA at 28 days after oral challenge was c. 4.7 log_10_ CFU/g compared to c. 7.7 log_10_ CFU/g in the mock-vaccinated group, but marked variability between individuals was observed with colonisation levels ranging from 3.5 to 8 log_10_ CFU/g (*p* ≥ 0.05; Figure 3B).

In vaccinated chickens challenged by exposure to pre-colonised seeder-birds, we observed no statistically significant differences between the groups at either time post-challenge (Figure 3C). Inter-animal variation in caecal colonisation was greater than observed during the development of the natural transmission model (Figure 2B), but colonisation following vaccination was evaluated in birds aged 35 or 56 days old rather than 28 or 31 days old as in the pilot study. Group sizes were chosen as power calculations based on variance observed in preceding studies indicated they could detect a difference of 2 log_10_ CFU/g with 80% power within the 95% confidence limit.

### 3.4. Antigen-Specific Humoral Immune Responses

Antigen-specific Ig in the sera of chickens was quantified by ELISA using the unglycosylated or glycosylated forms of FlpA as the capture antigen. Using both capture antigens, significantly elevated vaccine-specific Ig levels were detected in chickens vaccinated with FlpA or FlpA-10×GT at day 28 (2 weeks after the second vaccination), but not at day 14 after vaccination on the day of hatch (Figure 4A,B). The magnitude of antigen-specific Ig responses was similar whether FlpA or FlpA-10×GT were used as the capture antigen, indicating a limited response to the *N*-glycan per se, at least within the limit of sensitivity of the assay. FlpA-specific immune responses were comparable and elevated in all groups at 7 days post-challenge, presumably owing to seroconversion against FlpA following *C. jejuni* infection (Figure 4A,B). Minimal cross-reactivity was observed when serum from FlpA and FlpA-10×GT vaccinated chickens was tested against captured 6 × His-tagged NetB, purified from *E. coli* in the same way (Figure 4C). This indicates that the responses induced by FlpA or FlpA-10×GT are specific to those proteins rather than contaminants co-purified from *E. coli*. Serum from FlpA and FlpA-10×GT-vaccinated chickens was also tested against NetB-10×GT to determine if *N*-glycan-specific responses occurred. The serum from birds vaccinated with FlpA-10×GT reacted with NetB-10×GT to a significantly greater extent than serum from mock-vaccinated birds (Figure 4D), although this was only observed after both sets of vaccinated birds had been challenged with *C. jejuni*.

No significant correlation was detected between levels of FlpA or FlpA-10×GT serum Ig and caecal colonisation in the cognate chickens (Appendix A).

## 4. Discussion

Handling or consumption of poultry meat is a key risk factor for human campylobacteriosis. Other laboratories have reported that protection against *C. jejuni* colonisation can be conferred by vaccination of chickens with *Corynebacterium diphtheriae* ToxC with up to nine *N*-linked *C. jejuni* heptasaccharide glycan moieties [14], a live-attenuated *E. coli* strain engineered to express the *C. jejuni* heptasaccharide glycan on its surface [14], or outer membrane vesicles of *E. coli* decorated with the *C. jejuni* glycan [16]. Building on these findings, we evaluated a subunit vaccine based on the fibronectin-binding adhesin FlpA with up to ten *N*-glycan moieties in chickens. An internal peptide of FlpA has previously been reported to confer protection against caecal *C. jejuni* colonisation in broilers [19]. However, we previously reported that vaccination of chickens with full-length FlpA or FlpA with up to two *N*-glycan moieties was not protective in our model [18]. As the glycan dose was estimated to be markedly lower when FlpA-2×GT was used relative to the study using ToxC with up to nine *N*-glycans [14], we reasoned that protection may be improved by modifying FlpA such that up to ten *N*-glycan moieties are present. Relative glycan dose can considerably increase the efficacy of PGCT-based vaccines, as shown by increasing O-antigen occupancy from 2 to 10 moieties in a *Francisella tularensis* glycoconjugate vaccine [20]. Further, to address the hypothesis that oral inoculation may overwhelm vaccine-induced immune responses, we developed a model to mimic the natural transmission of *Campylobacter* based on the co-housing of vaccinated birds with seeder-birds pre-colonised by *C. jejuni*. Such a model was used to evaluate the impact of transplantation of a mature caecal microbiome to neonatal chicks against *C. jejuni* colonisation compared to protection against direct challenge by oral gavage. While microbiota transplantation was protective in both cases, the effect was much more significant when the seeder-bird model of natural transmission was used [23].

While vaccination of chickens with FlpA or FlpA-10×GT induced significant antigen-specific serum antibody responses, no statistically significant reductions in caecal *C. jejuni* colonisation were observed. Moreover, we were unable to detect a significant increase in antigen-specific IgY when FlpA-10×GT was used as a capture antigen relative to unglycosylated FlpA when analysing serum from chickens vaccinated with FlpA-10×GT, indicating that glycan-specific responses are limited or not detected owing to the sensitivity of the assay. A slight but significant antibody response against the *N*-glycan was detected when NetB-10×GT was used as a coating antigen. and serum from FlpA-10×GT-vaccinated birds was used, relative to that of mock-vaccinated birds, although this was only evident after both vaccination and challenge.

Nothaft et al observed marked variation between birds in the level of protection conferred by an attenuated *E. coli* live vaccine expressing the *C. jejuni N*-glycan on its surface, and no difference in vaccine-specific serum IgY was observed between responders and non-responders [14]. The role of antibodies in protection therefore remains unclear, although it may be that serum Ig levels do not adequately reflect mucosal IgA in either the magnitude or specificity of responses. Chemical bursectomy by cyclophosphamide treatment of chicks, which primarily affects the B lymphocyte compartment, has been reported to reduce clearance of *C. jejuni* from the jejunum and ileum of birds challenged at 3 weeks of age and sampled up to 7 weeks of age, although no effect was detected in the caecum [24]. In studies of a longer duration, B cells and secretory IgA were associated with control of *C. jejuni* in the caeca by 9 weeks post-infection, although a cyclophosphamide-sensitive non-B cell compartment may be involved [24]. It is possible that protective effects associated with FlpA/FlpA-10×GT vaccination may have been detected had sampling been continued beyond 4 weeks post-challenge; however, such findings would need to be considered in the context of the objective of vaccinating broilers, which frequently enter the food system from 5–6 weeks of age. Studies with edited chickens lacking the Ig heavy chain [25] could formally establish the role of antibodies in any protection conferred by *C. jejuni* vaccines.

The glycan dose per FlpA-10×GT vaccination in the current study was estimated to be 9.98 μg. This compares to 15 μg for the ToxC-9×GT vaccine [14], 10 μg for ExoA-10×GT [15], and 3 μg for FlpA-2×GT [18]. They found the activity of the PglB oligosaccharyltransferase to be temperature-sensitive and optimal under these conditions [22]. Further refinements to optimise PglB activity and the expression of sugars required for glycan assembly may enhance the efficiency of glycosylation.

The reasons for the disparity between our findings and those of other authors when testing similar vaccines remain unclear. It has been reported that the composition of the microbiota is associated with the response of chickens to an *N*-glycan-decorated live *E. coli* vaccine, in particular operational taxonomic units classified as *Clostridium* spp., *Ruminococcaceae,* and *Lachnospiraceae* [26]. *Anaerosporobacter mobilis* is also associated with the responder phenotype, and studies have shown that co-administration of this strain and the *E. coli N*-glycan vaccine can enhance protection [26]. Similar findings were observed when co-administering the vaccine with *Lactobacillus reuteri*, a commonly used probiotic [27]. It is therefore plausible that differences in the gut microbiota of the birds used may explain differences in protection. Further, it is possible that the microbiota of chickens used in our studies contains organisms that display glycans similar in structure to the *C. jejuni N*-glycan, such that immunological tolerance limits adaptive responses to the *N*-glycan. Differences also exist in the vaccination regimen and *C. jejuni* strain chosen, reinforcing the importance of attempting to standardise variables across laboratories where feasible to permit direct comparison.

The absence of protection by glycoconjugate vaccines in our studies can also be interpreted in the context of the variation seen for similar vaccines tested by independent laboratories. For example, Wyszyńska et al. reported a 6 log_10_ CFU/g reduction in *C. jejuni* colonisation of chickens vaccinated with an *S*. Typhimurium Δ*crp*Δ*cya* mutant expressing *C. jejuni* CjaA [28], yet an *S*. Typhimurium Δ*aroA* vaccine vectoring CjaA reduced caecal *C. jejuni* levels by 1.4 log_10_ CFU/g [29]. In the latter study, six independent trials of the same design were conducted, and, in some replicates, negligible protection was detected, while in others the reduction in caecal colonisation was closer to 3 log_10_ CFU/g. Others have similarly been unable to replicate protection with attenuated *Salmonella* vectoring CjaA [30] or observed a modest protective effect [31]. Such studies reinforce the importance of research to reproduce the findings of others and of publishing data that may be perceived to be negative.

Previous research has shown that the structure of *C. jejuni* populations in chickens is highly dynamic and unpredictable, such that findings from a single study with a small number of birds are to be interpreted with caution. This may reflect phase-variable gene expression, which generates variants that may emerge under selection. However, this does not appear to explain the non-responder phenotype in chickens vaccinated with *N*-glycan-decorated *E. coli,* as analysis of poly(G)-containing loci of the input strain compared to *C. jejuni* isolates from non-responder birds found no evidence of phase state selection [27]. Recent studies have shown that after immunisation of mice with the lipopolysaccharide O-antigen of *Salmonella enterica* serovar Typhimurium, *Salmonella* can rapidly escape vaccine-induced responses via mutations in biosynthetic pathways or by phase variation [32]. By immunising with all four possible variants of the O-antigen, including those modified by acetylation of abequose residues or glucosylation of galactose residues, escape was found to be much more difficult and typically came at a fitness cost to the bacteria [32]. It remains to be seen if *C. jejuni* can adapt in this way, and if so, whether this is associated with the variation in protection observed between vaccination studies.

## Figures and Tables

**Figure 1 vaccines-12-00395-f001:**
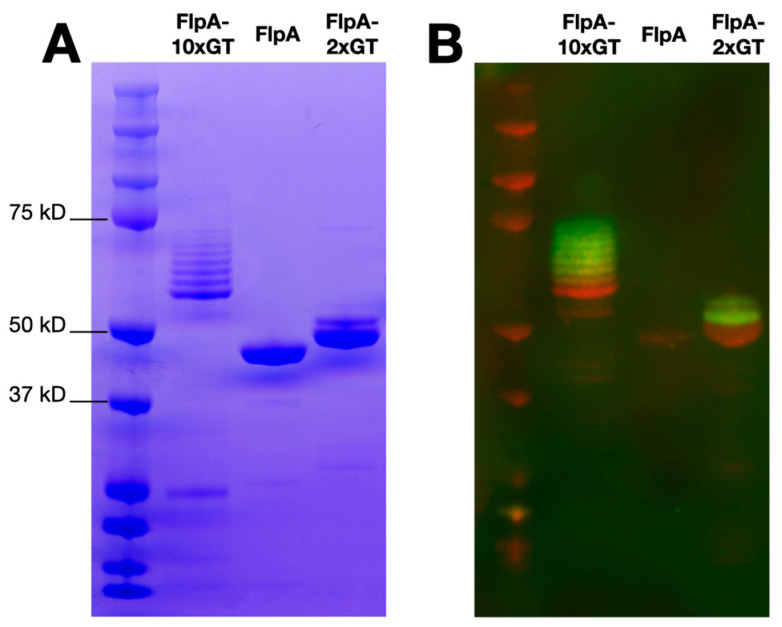
Purity and glycosylation of vaccine antigens. Panel (**A**) shows SDS-PAGE analysis of affinity purified FlpA, FlpA-2×GT, and FlpA-10×GT with proteins of the expected size. For FlpA-2×GT and FlpA-10×GT, species of a higher molecular weight correspond to *N*-glycosylated variants, as evident from staining with biotinylated soybean agglutinin detected with a streptavidin fluorophore conjugate as shown in panel (**B**) (green). The proteins were detected in panel (**B**) using mouse monoclonal anti-6xHis conjugated to the 680LT fluorophore (red). Original, uncropped images can be found in the Appendix A.

**Figure 2 vaccines-12-00395-f002:**
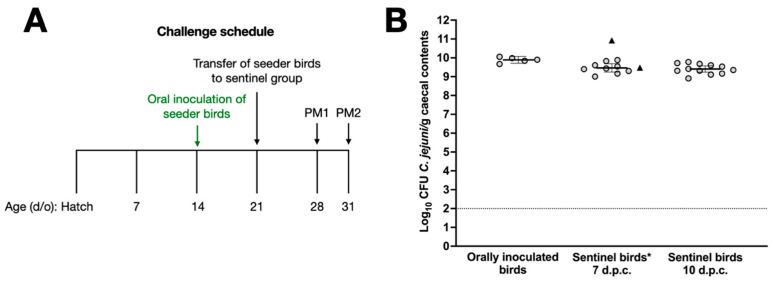
Validation of a model to mimic the natural transmission of *C. jejuni* between chickens. Panel (**A**) shows the schedule for oral gavage of seeder-birds with 10^5^ *C. jejuni* NCTC 11168H, followed by the introduction of 2 seeder-birds into a group of 23 naïve sentinels. Caecal colonisation by *C. jejuni* NCTC 11168H in sentinel birds at 7 and 10 days after introduction of the seeder-birds is shown in panel (**B**), together with evidence that seeder-birds were reliably colonised. Days post-challenge are abbreviated to d.p.c. * Bacterial counts of seeder-birds are shown as triangles and were excluded from the statistical analysis.

**Figure 3 vaccines-12-00395-f003:**
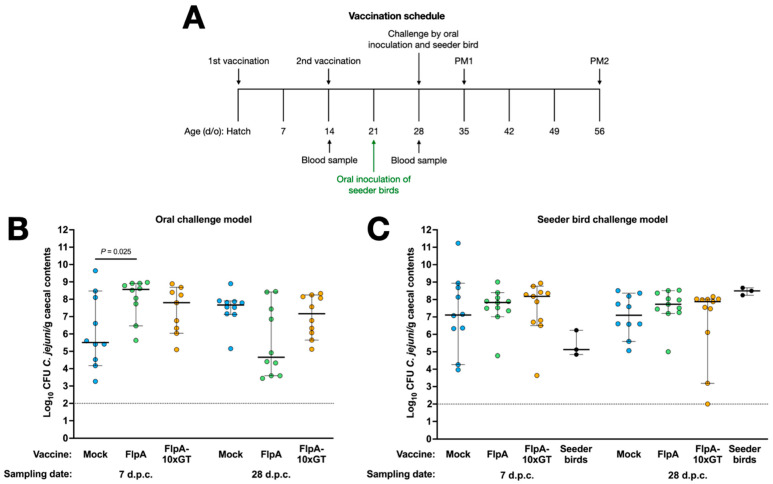
Vaccination of chickens with FlpA or FlpA-10×GT does not significantly reduce caecal *C. jejuni* colonisation. Panel (**A**) shows the schedule of vaccinations, challenge, blood sampling, and postmortem examination. Caecal colonisation is shown for groups vaccinated with FlpA, FlpA-10×GT, or control and challenged by oral inoculation of 10^5^ CFU of NCTC 11168H (panel (**B**)) or by exposure to pre-colonised seeder-birds (panel (**C**)). Days post-challenge are abbreviated to d.p.c.

**Figure 4 vaccines-12-00395-f004:**
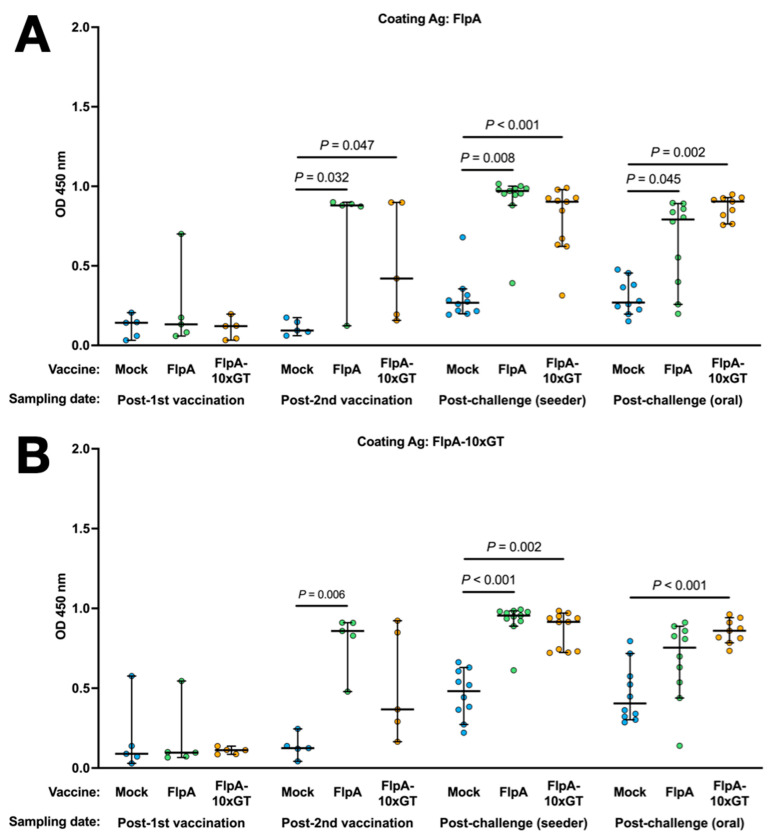
Quantification of antigen-specific Ig responses following vaccination and challenge by ELISA. The coating antigens used were unglycosylated FlpA (panel (**A**)), FlpA-10×GT (panel (**B**)), unglycosylated NetB (panel (**C**)), or NetB-10×GT (panel (**D**)). These were screened against 1:50 dilutions of serum from chickens that were mock vaccinated or vaccincated with FlpA or FlpA-10×GT, collected 14 days after the first vaccination on the day of hatch or 14 days after the second vaccination (day 28 of age). Serum was also analysed from birds 7 days after the *C. jejuni* challenge. The data are shown separately for birds that were challenged by oral gavage or by contact with seeder-birds. Significant differences between groups are indicated, and *p* values are indicated.

## Data Availability

Original images can be found in the Appendix A. Original numerical data underlying other figures and tables are available from the corresponding author upon request.

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
