# Peer review of "Evaluation of a FlpA Glycoconjugate Vaccine with Ten N-Heptasaccharide Glycan Moieties to reduce Campylobacter jejuni Colonisation in Chickens"

_vaccines, 2024, doi:10.3390/vaccines12040395_

Round 1

Reviewer 1 Report

Comments and Suggestions for Authors

Corona-Torres et al describe the unsuccessful experiment using a heptasaccaride FlpA glycoconjugate vaccine against C. jejuni.

The reviewer considers it very important to describe non – successful experiments as these can potentially teach more than successful ones.

The reviewer is not aware as to how a vaccine given subcutaneously should be effective in the gastrointestinal (GI) tract of chickens. C. jejuni is usually not disseminated systemically but remains in the GI tract. It is well known that IgY is present in large amounts in eggs but evidence is unknown to the reviewer that chickens can secrete IgY into the GI tract or vaccinated via the GI tract. Moreover, there is evidence that IgY is rapidly degraded in the GI tract (see Wang et al., 2021). This is in stark contrast to dimeric IgA in mammals.

So what is really the rational to vaccinate chickens against GI restricted bacterial infections ? 

1.       Wang H, Zeng X, Lin J. 2021. Ex Vivo Evaluation of Egg Yolk IgY Degradation in Chicken Gastrointestinal Tract. Front Immunol 12:746831.

Author Response

We thank the reviewer for their positive assessment of our manuscript. While the study was unsuccessful in the sense that vaccination with FlpA-10xGT did not elicit protection against intestinal colonisation by C. jejuni following in-contact exposure or oral dosing, we detected both FlpA- and N-glycan-specific Ig responses. That these were not protective balances findings from other laboratories, and adds to literature on the feasibility of achieving robust vaccine-mediated protection against Campylobacter in poultry.

Regarding the feasibility of inducing secretory IgA responses following parenteral vaccination, it is well established that this occurs in mammals (Clements JD & Freytag LC. 2016. Parenteral vaccination can be an effective means of inducing protective mucosal responses. Clin Vaccine Immunol 23:438-441), and chickens are similarly able to switch the Ig heavy chain (e.g. from IgM to IgA) via recombination in B cells without affecting the epitope specificity of the variable domain. Such IgA secreting cells are abundant in avian gut-associated lymphoid tissue. Indeed, it has been reported that Campylobacter outer membrane proteins encapsulated in nanoparticles elicited higher sIgA responses when administered subcutaneously relative to when they were given orally in chickens (Annamalai T, et al. 2013. Evaluation of nanoparticle-encapsulated outer membrane proteins for the control of Campylobacter jejuni colonization in chickens. Poult Sci 92:2201-11). We partly chose a parenteral route of administration as a glycoconjugate vaccine for Campylobacter was reported to be protective in chickens when given intramuscularly (Nothaft H, et al. 2016. Engineering the Campylobacter jejuni N-glycan to create an effective chicken vaccine. Sci Rep 6:26511).

Commercial vaccines for control of gastrointestinal pathogens in poultry are frequently delivered via intramuscular or subcutaneous routes, as with inactivated Salmonella vaccines (e.g. Zoetis PoulVac SE, MSD Animal Health Salenvac T/E, and Ceva CORYMUNE) and Salmonella-specific sIgA has been detected after subcutaneous vaccination of chickens with such bacterins (Ishida Y, et al. 2018. Induction of mucosal humoral immunity by subcutaneous injection of an oil-emulsion vaccine against Salmonella enterica subsp. enterica serovar Enteritidis in chickens. Food Saf (Tokyo). 6:151-155). It is recognised that antibody may be degraded in the gut, but we note that the reference cited  by the reviewer relates to orally administered IgY that will pass through gut compartments where it may be digested, whereas IgA can be secreted from mucosal surfaces at locations where Campylobacter persists where proteolysis is less likely.

Reviewer 2 Report

Comments and Suggestions for Authors

This study is well-planned, methods are proper and sufficient. The results reflect a carefully executed study. The discussion section is detailed and clear; the manuscript is well-written. I believe that the manuscript is of interest for the readers of the journal.

I have only minor comments.

1.       Abstracte: The first sentence contains a false statement. Campylobacter is not a major cause of acute gastroenteritis in humans. It should be corrected, especially since the authors correctly indicate the Campylobacter contribution to gastroenteritis in the introduction.

2.       The data described in the Discussion (lines 344-352) repeats what has already been described in the Report (lines 70-81). It's worth removing the repeats.

Author Response

We thank the reviewer for their positive assessment of our manuscript. The first sentence of the abstract stated ‘Campylobacter is a major cause of acute gastroenteritis in humans and infections can be followed by inflammatory neuropathies and other sequelae’. We consider this to be true and is supported by global estimates of the burden of foodborne illness (e.g. Havelaar AH, et al. 2015. World Health Organization global estimates and regional comparisons of the burden of foodborne disease in 2010. PLoS Med 12:e1001923). The abstract has not been revised as a consequence. With regard to repetition, we consider it important to preface both our study, and discussion of our findings, with the observations from other laboratories reporting that protection against C. jejuni colonisation can be conferred by N-glycan decorated E. coli, outer membrane vesicles and subunit vaccines. We have not observed such protection in our studies, and wish to describe the rationale of our study, and disparity in findings, in this context.

Reviewer 3 Report

Comments and Suggestions for Authors

The manuscript is interesting considering the importance of controlling Campylobacter jejuni in poultry production since it is the major cause of gastroenteritis in humans derived from the consumption of contaminated food. However, there are some questions and comments that the authors must address before publishing the manuscript.

L.47. However, in some broiler lines, C. jejuni....

L.155. At what stage of bacterial growth kinetics was C. jejuni when the birds were inoculated?

L.173. Change "killed" to "euthanized"

L.191. 2 seeder-birds per group? Were the experimental groups together or separate?

L.200. Change "following" to "after"

L.219. (BioTek Instruments, city, country)

L.225. Delete "of"

L.279. This may be due to the stage in which the bacteria were in their growth kinetics, for this reason it was previously asked if the authors knew the stage in which they were when the birds were challenged.

L.320. Delete "and"

Author Response

We thank the reviewer for their thorough and positive assessment of our manuscript. The requested changes have been made in the revised version. We have specified the growth phase of the bacteria used for inoculation and updated details of the suppliers of reagents or equipment. 

Round 2

Reviewer 1 Report

Comments and Suggestions for Authors

This is fine

Author Response

No response is required. Reviewer 1 states that are response to issues raised in Round 1 of review is fine.